# Anatomy, Biomechanics, and Reconstruction of the Anterolateral Ligament of the Knee Joint

**DOI:** 10.3390/medicina58060786

**Published:** 2022-06-10

**Authors:** Jun-Gu Park, Seung-Beom Han, Chul-Soo Lee, Ok Hee Jeon, Ki-Mo Jang

**Affiliations:** 1Department of Orthopaedic Surgery, Anam Hospital, Korea University College of Medicine, Seoul 02841, Korea; jgpark11@gmail.com (J.-G.P.); sbhan1107@gmail.com (S.-B.H.); anotherlife@naver.com (C.-S.L.); 2Department of Biomedical Sciences, Korea University College of Medicine, Seoul 02841, Korea; ojeon@korea.ac.kr; 3Department of Sports Medical Center, Anam Hospital, Korea University College of Medicine, Seoul 02841, Korea

**Keywords:** knee joint, anterolateral ligament, anterolateral ligament reconstruction, lateral extra-articular tenodesis, anterior cruciate ligament, anterior cruciate ligament reconstruction

## Abstract

Despite remarkable advances in the clinical outcomes after anterior cruciate ligament reconstructions (ACLRs), residual rotational instability of the knee joint remains a major concern. Since the anterolateral ligament (ALL) on the knee joint has been “rediscovered”, the role of anterolateral structures, including ALL and deep iliotibial band, as secondary stabilizers of anterolateral rotatory instability has gained interest. This interest has led to the resurgence of anterolateral procedures combined with ACLRs to restore rotational stability in patients with anterior cruciate ligament (ACL) deficiencies. However, the difference in concepts between anterolateral ligament reconstructions (ALLRs) as anatomical reconstruction and lateral extra-articular tenodesis (LETs) as non-anatomical reinforcement has been conflicting in present literature. This study aimed to review the anatomy and biomechanics of anterolateral structures, surgical techniques, and the clinical outcomes of anterolateral procedures, including LET and ALLR, in patients with ACL deficiencies.

## 1. Introduction

Tears and sprains in the anterior cruciate ligament (ACL) are common knee injuries in active patients. ACL reconstruction (ACLR), a well-established procedure, restores knee joint stability, improves function, and supports an eventual return to sports. Although the advances in surgical techniques and improved understanding of the ACL anatomy and biomechanics have improved clinical outcomes, residual rotatory instability (related to unsatisfactory clinical outcomes and risks of re-tear) was observed in 25% of the patients [1,2,3].

Historically, isolated lateral extra-articular tenodesis (LET) was performed to restore the anterolateral stability of knee joints before introducing the arthroscopic intra-articular ACLR [4]. However, after the development of arthroscopic ACLR, isolated LET was no longer performed primarily because of its invasive nature and unfavorable long-term clinical results [5].

Since the anterolateral ligament (ALL) was “rediscovered” as a distinct ligamentous structure on the anterolateral aspect of the knee joint, the anterolateral complex (ALC) of the knee joint, consisting of a superficial and deep capsule-osseous layer of the iliotibial band (ITB) and the ALL have gained attention [6,7]. Recently, several anatomic and biomechanical studies have demonstrated that ALCs of the knee joint potentially contribute to the anterolateral stability in ACL-deficient knees [8,9,10,11,12]. Thus, additional lateral extra-articular procedures, such as LET and anterolateral ligament reconstruction (ALLR), combined with intra-articular ACLR have been introduced for improved control over the anterolateral instability [13].

Although the biomechanical evidence suggests that renewed LET and ALLR combined with ACLR is a better option for the improvement of anterolateral instability compared with isolated ACLRs [14,15,16], the clinical applications of these procedures in terms of proper indication and superiority arising from their respective limitations are still debatable. In LET using ITB, a risk of over-constraint on the lateral compartment persists, and knee joint kinematics can be altered owing to its non-anatomical reconstruction feature. Inconsistent findings in terms of presence, anatomic location, and length changes of the ALL among several cadaveric studies have made the surgical technique in terms of tunnel placement and fixation angle of the graft widely debated [17,18,19,20].

Recently, the clinical advantages of ALLR combined with ACLR compared to isolated ACLR have been demonstrated in multiple studies: reduced re-tear rate, a lower reoperation rate for secondary meniscectomy, and a higher rate of return to sports [21,22,23]. However, the proper indications and optimal surgical techniques are still controversial, owing to conflicting indications and surgical techniques across studies. This article reviews the anatomy and biomechanics of ALL, and the current indications and clinical outcomes of anterolateral procedures, including LET and ALLR, in patients with ACL deficiency.

## 2. Anatomy of Anterolateral Complex and Anterolateral Ligament

The ACL is an important intra-articular ligament structure of the knee joint, which resists anterior tibial translation and rotational loads. The anterolateral aspect of the knee joint consists of multiple extra-articular structures which constitute the ALC, usually responsible for anterolateral stability. The ALC consists of the superficial ITB and iliopatellar band, the deep ITB (Kaplan fibers, retrograde condylar attachment continuous with the capsulo-osseous layer), and the ALL [6]. The ITB is connected to the distal femur through Kaplan fibers and has no attachment to the lateral femoral epicondyle (LFE). The ALL is a ligamentous structure clearly distinct from the ITB, and both ‘deep layer’ (Kaplan fibers) and ‘capsulo-osseous layer’ of the ITB should not be confused with the ALL. The ALL and lateral collateral ligament (LCL) are in proximity. However, the ALL is more superficial. The variations in reports on the prevalence of ALL have made the existence of ALL as a distinct ligamentous structure unclear. A recent systematic review reported the prevalence of ALL in 83.0% of patients [24]. Inconsistent findings in the femoral origin of ALL were reported in cadaveric studies [7,25,26]. Claes et al., described the origin of ALL as slightly anterior to the origin of LCL. Helito et al., described the femoral origin of ALL 2.2 mm anterior and 3.5 mm distal to the attachment of the LCL [26]. Dodds et al., described the origin as 8 mm proximal and 4.3 mm posterior from the LFE [25]. Recently conducted cadaveric studies have predominantly reported the constant femoral origin of ALL as posterior and proximal to LFE [20,24]. The tibial insertion of ALL showed relatively consistent location as halfway between Gerdy’s tubercle and the tip of the fibular head below 4.0 mm to 7.0 mm from the lateral joint line (Figure 1).

## 3. Biomechanics of Anterolateral Complex and Anterolateral Ligament

Several biomechanical studies have demonstrated that the ALL acts as a secondary stabilizer of anterolateral rotatory instability [10,14,27,28,29]. In the native knee joint, the ACL provides the primary restraint for anteroposterior and anterolateral rotatory stability. The load carried by the ALL when applying anteroposterior and tibial internal rotation forces is relatively insignificant compared with that of the ACL in native joints. Thein et al., found that the load carried by the ALL in intact ACL knees was 16.6 N in a simulated pivot shift test at 30° of knee flexion, corresponding to 16.6% of the load on the ACL. However, after the ACL was sectioned, the load carried by the ALL increased three-folds to 54.7 N [30], suggesting that ALL’s role might increase in ACL deficiencies. Several cadaveric studies investigated the role of the ALL on rotatory instability in ACL-deficiencies by sequential sectioning of the ACL and ALL. The findings suggested that further resection of the ALL in ACL-deficient knees increased rotatory instability [28]. Inderhaug et al., reported that the addition of anterolateral lesions to ACL-deficient knees significantly increased knee laxity from ACL-deficient states [16]. The extent of rotatory instability in ALL deficiency was more obvious for increased knee flexion angles. At early flexion angles between 0° and 30°, the ACL acted as the main stabilizer for both anteroposterior translation and tibial internal rotation. The ACL limited internal rotation near full extension, where it was tight [31]. However, at degrees of flexion exceeding 30°, the ACL’s contributions to rotatory stability decreases, and ALL’s role increases. Parsons et al., reported that the ALL’s contribution to stability in internal rotation significantly increased between 0° and 90° of knee flexion [8]. Nitri et al., compared the tibial internal rotation after combined ACLR and ALLR and only ACLR in both ACL-and ALL-deficient knees. They found that the tibial internal rotation increased in accordance with increasing flex-ion angles in ALL-deficient knees compared with intact knees [14] (Figure 2).

Biomechanical studies demonstrated that anatomical single bundle ACLR could not restore normal knee kinematics, with increased retention of rotatory laxity [32]. Furthermore, in ACL-and ALL-deficient knees, isolated ACLRs resulted in a significant increase in residual rotatory instability. Nitri et al., reported that isolated ACLRs showed significant residual rotatory instability compared with combined ACLR and ALLR in cases of ALL and ACL deficiency [14].

There are conflicting views regarding the essential ALC component for rotary stability, especially with respect to the deep ITB (capsulo-osseous layer) and ALL. Some authors have suggested that the ALL contributes less than the deep ITB. Kittl et al., reported that in ACL deficiencies, the superficial ITB’s contribution was higher to rotatory stability at higher flexion angles, and the deep ITB predominantly contributed at lower flexion angles. The proportion of ALL was relatively small (10%). They suggested that the deep ITB is more important for rotatory stabilization in knees with ACL deficiencies [9]. However, Sonnery-Cottet et al., demonstrated that, even after sectioning of the ITB, an additional ALL section resulted in a significant increase in internal rotation [28]. Similar findings were reported by Ahn et al., wherein the ALL section in ACL-deficient knees significantly increased tibial internal rotation, with preserved Kaplan fibers between the femur and ITB [33]. Geeslin et al., compared the rotatory stability between the resection of ALL and Kaplan fibers in ACL deficiencies, finding no significant difference in the extent of increased internal rotation, using simulated pivot shift test at 15° and 30°. However, at higher flexion (>60°), the section of Kaplan fibers was capable of greater tibial internal rotation than the ALL section [11].

This inconsistency might be because of heterogeneity in dissection techniques, testing conditions, and the cadaver condition. Additionally, the difference in the contribution of rotatory stability by the ITB and ALL at knee flexion >60°, demonstrated previously in biomechanical studies using a sectioned knee cadaver, may not be significant in the native knee joint. The pathologic residual instability, which presented with subluxation and reduction of the tibia in the pivot shift phenomenon, occurs between 30° and 60° of knee flexion, and, at angles exceeding 60°, the tibia is reduced by ITB in the native joint [34]. However, in the sectioned knee cadaver, the reduction of the tibia by ITB disappeared, owing to disrupted proximal attachment of the ITB. Moreover, given the complexity of the anterolateral structures of the knee, the rotatory in-stability cannot be adequately explained using a single structure. Therefore, further investigations based on standardized methods that mimic the native knee joint are required. Regardless of this conflict, previous biomechanical studies have consistently demonstrated that isolated ACLRs result in residual rotatory instability in knees with ACL and ALC deficiencies between 30° and 60° of knee flexion. This finding suggests that additional anterolateral procedures to control rotatory instability should be performed. A summary of recent biomechanical studies on anterolateral complex and anterolateral ligament is presented in Table 1.

Changes in the ALL length during knee flexion are important for determining the graft fixation angles during ALLRs. However, the length change patterns of the ALL during knee flexion vary across studies, demonstrating the non-isometric features of the ALL [7,25,26,36]. Some authors reported that the ALL length gradually increased during knee flexion. Claes et al., reported that the ALL length was 38.5 mm in extension and 41.5 mm in 90° of flexion [7]. Helito et al., reported that the ALL length increased by 16.7% from full extension to 90° of flexion [26]. In contrast, Dodds et al., found that the ALL was close to isometric between 0° and 60° and lax in 90° of flexion [25]. However, the descriptions of femoral insertion varied across these studies. Claes et al., and Helito et al., described femoral insertions as distal and anterior to the LFE, and Dodds et al., described as proximal to posterior to the LFE. Imbert et al., investigated the length changes in the ALL according to three different femoral insertions: at the center of the LFE, distal and anterior to the LFE, and proximal and posterior to the LFE, using a navigation system. With femoral insertions performed proximal and posterior to the LFE, the ALL increased in extension and decreased in flexion. Conversely, with femoral insertions distal and anterior to the LFE, and at the center of the LFE, the ALL decreased in flexion and increased in extension [36] (Table 2).

## 4. Diagnosis of Anterolateral Ligament Injury

Identification and evaluation of ALL injuries are crucial to predict the residual anterolateral rotary instability and determine the additional anterolateral procedures in ACL deficient knees. Although a positive pivot shift test is the most important clinical finding for additional anterolateral procedures, there is a limitation due to varying results depending on the experience and technical skills of the examiner [37]. Among several diagnostic methods, magnetic resonance imaging (MRI) is the most useful imaging tool for evaluating the ALL and combined pathologies. The detection rate of ALL on MRI scans is reported as high as around 80% to 100% [38,39]. The reported prevalence of ALL lesions ranges from 64% to 88% of ACL-injured knees [39,40,41]. However, the identification of the entire ALL is limited in the routine MRI protocols due to its thin structure, oblique course, and complex relationship among adjacent structures. A previous study reported that at least one portion of ALL was visualized on almost MRI scans (97.4%), whereas only two-thirds of cases characterized ALL entirely [38]. Moreover, the accuracy of discrimination in the severity and location of ALL lesions is reported to be relatively low [42]. More accurate identification of ALL lesions could be accomplished with standardized protocols. The most useful sequence for identification of the ALL on MRI is a coronal cut of proton density [38]. The ALL is identified as a thin and linear structure surrounded by fatty tissue and its meniscal portion is found just proximal to the inferior genicular artery, which is visualized as a dot on the coronal MRI images [43]. As a reference to the ALL on the coronal plane, the anatomy of ALL can be identified in more detail on axial and sagittal images [38].

## 5. Lateral Extra-Articular Tenodesis

LET was first performed by Lemarie in 1967 to control anterolateral rotatory instability in ACL-deficient knees before the arthroscopic intra-articular ACLR technique was introduced [4]. Although the symptoms of instability improved in the short-term, isolated LETs without ACL reconstructions had several limitations and unsatisfactory long-term outcomes, including residual or recurrent laxity, the deterioration of subjective scores over time, stiffness, and degenerative changes of lateral compartments. Consequently, no advantages were reported over arthroscopic ACLRs [5]. After the introduction of arthroscopic ACLRs, isolated LETs were no longer used.

Recently, several biomechanical studies have demonstrated the restoration of rotatory stability by combining LETs with ACLRs in the cases of residual rotatory instability after isolated ACLRs [13]. Renewed interest in LETs has emerged based on these promising results. Lagae et al., found that isolated ACLRs performed for ACL and ALL deficiencies resulted in residual rotatory laxity compared to intact knees at 30–100°. However, LETs combined with ACLRs were insignificant compared with intact knees [35]. Similar findings were reported in a previous biomechanical study by Inderhaug et al. They found that the addition of LETs to ACLRs restored the native knee kinematics at time zero [16].

Several LET techniques have been introduced, and variations exist between authors [44]. However, the general concept throughout the techniques is consistent with the reinforcement of anterolateral structures to control rotatory instability and share the load on ACL grafts. Among these techniques, the modified Lemaire technique is the most prevalent. It was modified from the original technique to decrease invasiveness. A brief description of the surgical procedure is presented below. A 5- to 6-cm incision is made just posterior to the LFE, the central slip of the ITB dimensioned 1 cm width and 8 cm length is harvested, maintaining distal attachment of the ITB on Gerdy’s tubercle. The stitched proximal end of the ITB graft is routed deep into the LCL and fixed just proximal and posterior to the LFE. Fixation is performed with the knee at 60–70° flexion and neutral rotation, applying 20-N tension [45].

Because LET is a non-anatomical reinforcement procedure, the main concern is its potential to over-constrain the lateral compartment, causing stiffness or accelerated development of lateral compartment osteoarthritis (OA). Some biomechanical studies have shown that the addition of LET to ACLR induced the over-restriction of tibial internal rotation compared to the intact knee [16,46]. Specifically, Inderhaug et al., suggested that the constraint on knee kinematics is affected by graft tension and knee flexion angle during fixation. They found that 40-N of graft tension resulted in the over-restriction of physiological internal rotation compared to 20-N of graft tension. Additionally, the fixation position was modified to the neutral rotation position, unlike the excessive external rotation position in the initial description of Lemaire [45].

The ITB graft’s orientation also affects the joint kinematics in the lateral compartment. In a cadaveric biomechanical study that investigated the effects of the ITB graft orientation between superficial and deep to the LCL, Inderhaug et al., found that the superficial orientation of the ITB graft over-constrained internal rotation, whereas the deep orientation of the ITB restored rotational kinematics to the intact state [16]. Some authors suggested that when the graft passes deep to the LCL, the LCL acts as a fulcrum to avoid over-constraint [16,47]. Moreover, Kittl et al., found that fixation on the posterior and proximal to the LE and passage of the graft deep to the LCL provide minimal length change throughout the knee joint motion [47].

Concerns related to the over-constraint of the lateral compartment remain an issue. However, a recent systematic review demonstrated insufficient evidence that LETs combined with ACLRs accelerated the lateral compartment OA in the long-term [48]. Further long-term clinical studies based on optimal tension and fixation angles should be conducted to reveal the influence of LET on the lateral compartment.

## 6. Anterolateral Ligament Reconstruction

Several biomechanical studies demonstrated improved rotational instability and restoration of normal knee kinematics through ALLRs combined with ACLRs in ALL and ACL deficiencies [16,44,49]. Although variations in ALLR surgical techniques exist in terms of femoral and tibial tunnel locations, graft types, fixation angles, and graft tension, the general concept is the anatomical reconstruction of the ALL. ALLRs are less invasive than LETs and can be performed with one or two small incisions on the femoral and tibial sides [50].

As demonstrated in a recent systematic review, the femoral tunnel of the ALL is generally created proximal and posterior to the LFE, following the descriptions in current anatomical studies [21,24]. To create the femoral tunnel, the LFE is palpated and identified. A 15-mm incision is made just proximal to the LFE, and the LFE is identified followed by a pin insertion 8 mm proximal and 4 mm posterior to the LFE. The tibial tunnel is located 5–10 mm below the lateral joint line and halfway between the tip of the fibular head and Gerdy’s tubercle. One or two tibial tunnels can be used. However, the use of two tibial tunnels may be advantageous in terms of restoring the broad base of the anatomic ALL tibial footprint. After placing the guide pin on the tibial side, the length changes between the femoral and tibial guide pin should be examined using suture tape during knee joint motion. The suture should be tight during extension to provide rotatory stability and slackened during 90° flexion to permit physiological internal rotation [36,50].

As aforementioned, regardless of the surgical technique, the femoral tunnel position should be fixed at full extension or slight flexion. Therefore, the ALL graft is tightened near knee extension to control the pivot-shift and slackened at 90° flexion to permit physiological internal rotation. For example, when the femoral tunnel is located distal and anterior to the LE, fixation in extension causes over-constraint in flexion [50].

In a recent systematic review, the semitendinosus and gracilis tendons were commonly used in ALLRs. The ultimate load to failure of the ALL was 50 N [51]. Considering that the ultimate load for failure of semitendinosus and gracilis tendons are 1216 N and 838 N, respectively, the semitendinosus and gracilis tendons provide sufficient strength for grafting in ALLRs [52].

However, the optimal graft tension remains controversial. Since the ALL is not a primary stabilizer, the minimal tension required for a check-rein effect may be sufficient rather than over-constrain. Previous biomechanical studies have investigated the restoration of internal rotation according to graft tension. Inderhaug et al., reported that ALLRs fixed with 20-N of tension at full extension restored normal knee kinematics [49]. Geeslin et al., showed similar findings for grafts carrying 20-N of tension [13]. In the biomechanical study with grafts fixed with 88-N at 70° of knee flexion, ALLRs excessively restricted the internal rotation at higher flexion angles [14]. Another study of ALLRs using a 6-mm single strand semitendinosus allograft tensioned with 88 N indicated overstrain during internal rotation [53]. Given the previous biomechanical studies, tension with 20-N would be optimal for restoring the rotatory stability without overstrain in combined ACLRs.

Another surgical issue in ALLR is the collision of femoral tunnels between the ACL and ALL. Unlike the Sonnery–Cottet technique, which uses the same femoral tunnel of the ALL and ACL [50], the independent femoral tunnel is at risk of violating each tunnel. When the ACL femoral tunnel is created using the outside-in method, the aperture on the lateral side of the two tunnels can be predicted. Thus, the collision risk is relatively low. However, in the inside-out femoral technique, the ACL femoral tunnel’s precise aperture on the lateral side is unpredictable. Stordeur et al., simulated numerous combinations of two tunnels to define the optimal position of the femoral tunnel. They suggested that the ACL femoral tunnel should be directed towards the posterior orientation and the ALL femoral tunnel should be aimed at one of the three following orientations: 40° axial and 10° coronal; 35° axial and 5° coronal; or 30° axial and 0° coronal [54].

The biomechanical superiority of combined ALLRS and ACLRs in controlling rotatory instability is undecided. A biomechanical study investigating the ability of ALLRs and LETs for restoring native knee kinematics found that both ALLR and LET improved knee stability during anterior translation and internal rotation in the knees deficient in ACL and ALL, without significant differences between the two techniques [15]. However, several comparative biomechanical studies between ALLR and LET combined with ACLR suggested that LET restricts rotatory instability more than ALLR. Conversely, it indicates that LET would further over-constrain the knee more than ALLR [44]. Neri et al., compared the knee kinematics in internal rotation between ALLR and several LET techniques. They fixed all grafts in neutral rotation at 30° of knee flexion and with 20 N of tension. In this biomechanical study, ACLR combined with ALLR restored normal knee kinematics in knees with ACL and ALL deficiency. However, the modified Lemaire technique induced over-constraint at a higher flexion angle [44]. A similar trend was reported by Geeslin et al. They found that both ALL and LET reduced the internal rotation more than the normal state, and a significantly greater reduction was noticed in LET, resulting in over-constraint [13].

The femoral insertion of ALLR and modified Lemaire technique share the same position proximal and posterior to the LFE. However, tibial insertions are different. In the modified Lemaire technique, the tibial insertion is the Gerdy’s tubercle. The tibial insertion of ALLR is halfway between Gerdy’s tubercle and the tip of the fibular head, resulting in a shorter lever arm compared to the modified Lemaire technique [44]. This biomechanical difference in the lever arm might induce additional restrictions in LET, compared with ALLR. However, additional long-term clinical studies are necessary for confirmation.

The indications for ALLRs remain controversial, with no current standardized guidelines established. However, a consensus has been reached that ALLRs should be considered in patients at high risk of re-tear, typically one of the following categories: (1) revision ACL reconstruction, (2) returning to competitive pivoting sports, (3) high-grade pivot shift test (more than grade 2), and (4) generalized ligamentous laxity [6].

## 7. Clinical Outcomes of Anterolateral Procedures

The primary goals of ALLR and LET combined with ACLRs are to improve instability symptoms and reduce the risk of graft tear. Therefore, several randomized clinical trials have investigated the additional advantages of anterolateral procedures combined with ACLRs compared with isolated ACLRs [55,56].

Getgood et al., compared the clinical outcomes of combined ACLRs and ALLRs with those of isolated ACLRs in a randomized clinical trial (RCT). They included 618 patients younger than 25 years displaying high-risk characteristics like high-grade pivot shift, return to pivoting sports, and generalized ligamentous laxity. With a two-year follow-up, the graft rupture rate of the combined ACLR and ALLR group was 4%, significantly lower than the 11% observed in the isolated ACLR group [55]. In a recent systematic review that included 7 RCTs, LETs showed improved stability, graft tear rate that was three times lower, and improved clinical outcomes compared with isolated ACLRs [57].

Like LET, RCTs and prospective cohort studies in ALLR also demonstrated improved stability, lower graft tear rate, and better clinical scores than isolated ACLRs [21,56,58]. Sonnery-Cottet et al., compared combined ALLRs and ACLRs performed in 221 patients with isolated ACLRs in 176 patients and reported a significantly lower graft tear rate in the combined ALLR and ACLR group at 4% compared to 11% in the isolated ACLR group [58]. Another RCT showed that residual laxity in term of the normalization of pivot-shift is lower in combined ALLRs and ACLRs than isolated ACLRs [56]. A recent systematic review demonstrated that the graft tear rate in combined ACLRs and ALLRs ranged between 2.7% and 11.1% and combined ACLRs and ALLRs had advantages over isolated ACLRs in terms of restoration of pivot shift [21].

In ACL-injured patients with generalized hyperlaxity, combined ACLRs and ALLRs had improved clinical outcomes compared with isolated ACLRs in terms of residual pivot shift and lower graft tear rate. Helito et al., evaluated 90 patients with generalized ligamentous laxity who underwent isolated ACLR (*n* = 60) or combined ACLR and ALLR (*n* = 30) and compared the graft rupture rates, residual laxity, and subjective clinical scores between the two groups. The graft tear rate was 3.3% in the combined ACLR and ALLR group, significantly lower than the isolated ACLR group (21.7%). Residual laxity was lower in the combined ACLR and ALLR group. However, the subjective clinical scores did not differ between the two groups [59].

The direct comparison of LET and ALLR combined with ACLR has not yet been performed by high quality clinical trials. However, anterolateral procedures demonstrated similar advantages over isolated ACLRs in terms of graft tear and restoration of residual rotary laxity. Despite comparable outcomes, the concepts of the procedures differed for the anatomical reconstruction in ALLRs and non-anatomical reinforcement in LETs. Therefore, the long-term effects of these procedures require further evaluation.

## 8. Conclusions

The ALL acts as a secondary stabilizer to the anterior cruciate ligament and helps resist internal knee rotation. Based on current literature, both ALLR and LET show improved pivot shift test and clinical outcomes compared to isolated ACLRs in patients with specific indications. These include high-grade pivot-shift, pivoting sports, generalized laxity, and revision surgery. The superiority of these two procedures remains uncertain. However, LET procedures may be associated with a risk of over-constraining the lateral compartment of the knee joint. Although several studies have demonstrated that combined ALLR and ACLR had improved postoperative clinical outcomes compared with isolated ACLR, further biomechanical and high-quality clinical investigations are needed to clarify the long-term effects of ALLR combined with ACLR in the patients with ACL deficiencies.

## Figures and Tables

**Figure 1 medicina-58-00786-f001:**
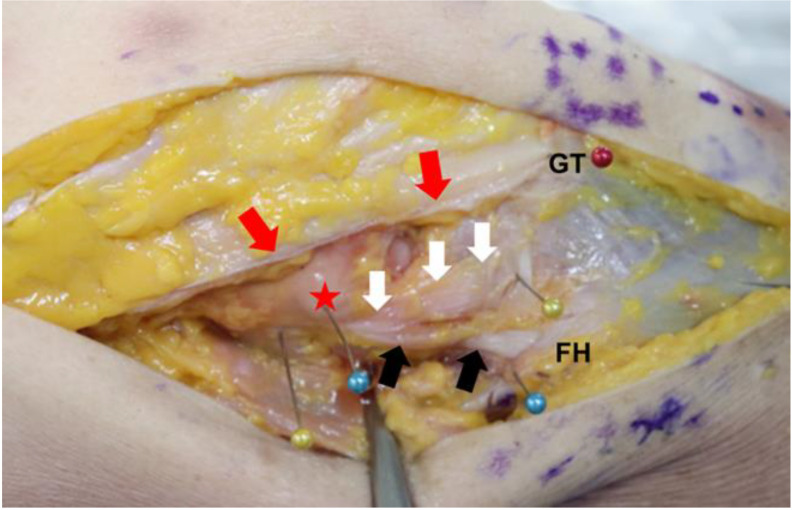
Anatomy of the anterolateral ligament and its surrounding structures. (white arrow, anterolateral ligament; black arrow, lateral collateral ligament; red arrow, iliotibial band (split); red asterisk, lateral femoral epicondyle; FH, fibular head; GT, Gerdy’s tubercle).

**Figure 2 medicina-58-00786-f002:**
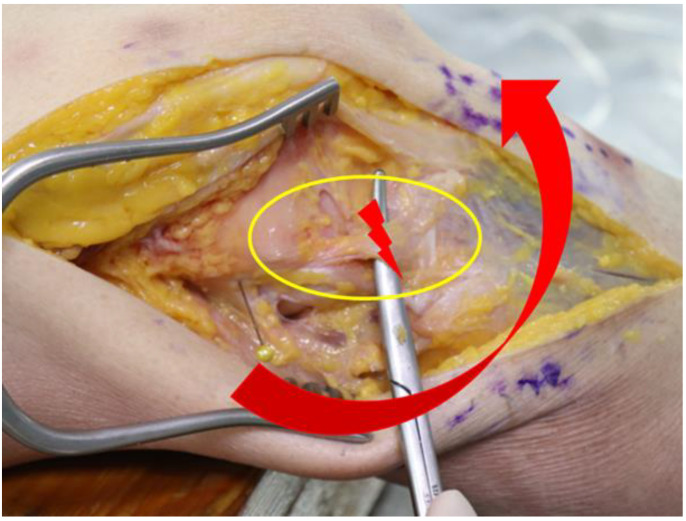
Resection of anterolateral ligament increases anterolateral rotatory instability in anterior cruciate ligament-deficient knees.

**Table 1 medicina-58-00786-t001:** Summary of biomechanical studies on anterolateral complex and anterolateral ligament of the knee joint.

Studies	Years	Specimens	Testing Conditions	Main Findings
Parsons et al. [8]	2015	Sectioned cadaveric knees	Evaluation of load on the ALL in tibial internal rotation according to flexion angle.	Contribution of the ALL to stability in internal rotation significantly increased between 0° and 90° of knee flexion
Thein et al. [30]	2016	Sectioned cadaveric knees	Comparison of load on the ligament in ACL and ALL deficiency.	In the ACL-intact knee, the load on ALL was minimal. In the ACL-deficiency knee, the load on ALL increased three-folds in the pivot shift-test.
Nitri et al. [14]	2016	Sectioned cadaveric knees	Comparison of amount of internal rotation between ALL deficiency and ALLR in ACLR	Increased tibial internal rotation in ALL deficiency
Kittl et al. [9]	2016	Sectioned cadaveric knees	Sequential resection of superficial ITB, deep ITB, ALL and ACL	1. From 0° to 30°, ACL was the primary restraint to internal rotation. 2. Superficial and deep ITB contributes over 50% of resistance to internal rotation above 30′ of flexion. 3. ALL and anterolateral capsule had a minor role in restraining internal rotation
Sonnery-Cottet et al. [28]	2016	Fresh-frozen cadaveric whole lower limbs	Sequential resection of ACL, ALL, ITB	In ACL or ITB deficiency, resection of ALL further increased tibial internal rotation.
Inderhaug et al. [16]	2017	Sectioned cadaveric knees	Sequential resection of ACL and anterolateral complex (ALL and deep ITB)	Additional resection of anterolateral complex increased tibial internal rotation in ACL deficiency. Restoration of native knee kinematics in LET combined with ACLR
Geeslin et al. [11]	2018	Sectioned cadaveric knees	Comparison of amount of internal rotation between resection of ALL and distal Kaplan fiber of ITB in ACL deficiency	Greater increased tibial internal rotation in resection of distal Kaplan fiber of ITB than ALL at higher flexion angle (60°–90°)
Geeslin et al. [13]	2018	Sectioned cadaveric knees	Comparison of residual internal laxity between LET and ALLR combined with ACLR in ACL and ALL deficiency knee	Increased residual internal laxity (up to 4°) in isolated ACLR Reduced residual internal laxity in both ALLR and LET (over-constraint in LET)
Lagae et al. [35]	2020	Sectioned cadaveric knees	Comparison of residual internal laxity with various settings in ACL and ALL deficiency knee	Residual internal laxity in isolated ACLR Reduced residual internal laxity in LET combined with ACLR
Ahn et al. [33]	2022	Sectioned cadaveric knees	Sequential resection of ACL, ALL, and anterolateral capsule	Even in the preservation of ITB, resection of ALL increase the internal rotation in ACL deficiency.

ACL, anterior cruciate ligament; ACLR, anterior cruciate ligament reconstruction; ALL, anterolateral ligament; ALLR, anterolateral ligament reconstruction; ITB, iliotibial band, LET, lateral extra-articular tenodesis.

**Table 2 medicina-58-00786-t002:** Summary of length change patterns of the anterolateral ligament during knee flexion in the biomechanical studies.

Studies	Years	Specimens	Femoral Origin	Tibial Origin	Length Changes
Claes et al. [7]	2013	Embalmed cadaver	Slight Anterior to LCL	Between GT and FH	38.5 ± 6.1 mm (0°) 41.5 ± 6.7 (90°)
Helito et al. [26]	2013	Unpaired cadaver knees with CT scans	2.2 mm Anterior and 3.5 mm distal to LCL	Between GT and FH	37.9 ± 5.3 mm (0°) 39.3 ± 5.4 mm (30°) 40.9 ± 5.4 mm (60°) 44.1 ± 6.4 mm (90°)
Dodds et al. [25]	2014	Fresh-frozen cadaveric knees	8 mm proximal and 4.3 mm posterior to LFE	Between GT and FH	Close to isometric from 0° to 60° flexion. (1.7 mm shortening) Shortening of 4.1 mm from 60° to 90° flexion
Imbert et al. [36]	2016	Fresh-frozen cadaveric whole lower limbs	Proximal and posterior to LFE	Between GT and FH	46 ± 6 mm (0°) 39 ± 2 mm (120°)

LCL, lateral collateral ligament; GT, Gerdy’s tubercle; FH, fibular head; LFE, lateral femoral epiphysis; CT, computed tomography.

## Data Availability

Not applicable.

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
