# Peer review of "Anatomy, Biomechanics, and Reconstruction of the Anterolateral Ligament of the Knee Joint"

_medicina, 2022, doi:10.3390/medicina58060786_

Round 1
Reviewer 1 Report
Dear Authors,
I reviewed with pleasure this review on the role of anterolateral ligament in knee stability.
I found the review accurate and addresses the main fields, from anatomy to the biomechanical role up to the various surgical techniques.
However, I think it would be appropriate to add a paragraph on instrumental diagnostics, with particular attention to the methods of recognition and the more frequent localization of lesions found in MRI.
Please cite this article in this paragraph:
- https://pubmed.ncbi.nlm.nih.gov/31872345/
- https://pubmed.ncbi.nlm.nih.gov/27056290/
- https://pubmed.ncbi.nlm.nih.gov/25085699/
Author Response
Dear Authors,
I reviewed with pleasure this review on the role of anterolateral ligament in knee stability.
I found the review accurate and addresses the main fields, from anatomy to the biomechanical role up to the various surgical techniques.
However, I think it would be appropriate to add a paragraph on instrumental diagnostics, with particular attention to the methods of recognition and the more frequent localization of lesions found in MRI.
Please cite this article in this paragraph:
https://pubmed.ncbi.nlm.nih.gov/31872345/
https://pubmed.ncbi.nlm.nih.gov/27056290/
https://pubmed.ncbi.nlm.nih.gov/25085699/
Author response) Thank you for your comments. We agree that it would be appropriate to add a paragraph on instrumental diagnostics, with particular attention to the methods of recognition and the more frequent localization of lesions found in MRI. We added a paragraph regarding diagnosis of ALL injuries according to your suggestion.
Author action) We added a paragraph according to your suggestion. We cited aforementioned articles in this paragraph (Line 188-209 in revised manuscript)
“Diagnosis of anterolateral ligament injury
Identification and evaluation of ALL injuries are crucial to predict the residual anterolateral rotary instability and determine the additional anterolateral procedures in ACL deficient knees. Although a positive pivot shift test is the most important clinical finding for additional anterolateral procedures, there is a limitation due to varying results depending on the experience and technical skills of the examiner [36]. Among several diagnostic methods, magnetic resonance imaging (MRI) is the most useful imaging tool for evaluating the ALL and combined pathologies. The detection rate of ALL on MRI scans is reported as high as around 80% to 100% [37, 38]. The reported prevalence of ALL lesions ranges from 64% to 88% of ACL-injured knees [38-40]. However, the identification of the entire ALL is limited in the routine MRI protocols due to its thin structure, oblique course, and complex relationship among adjacent structures. A previous study reported that at least one portion of ALL was visualized on almost MRI scans (97.4%), whereas only two-thirds of cases characterized ALL entirely [37]. Moreover, the accuracy of discrimination in the severity and location of ALL lesions is reported to be relatively low [41]. More accurate identification of ALL lesions could be accomplished with standardized protocols. The most useful sequence for identification of the ALL on MRI is a coronal cut of proton density [37]. The ALL is identified as a thin and linear structure surrounded by fatty tissue and its meniscal portion is found just proximal to the inferior genicular artery, which is visualized as a dot on the coronal MRI images [42]. As a reference to the ALL on the coronal plane, the anatomy of ALL can be identified in more detail on axial and sagittal images [37].”
Reviewer 2 Report
This review is a well-written and novel documented on the topic of anterolateral ligament reconstruction and not only. The references of the article are new and the experts in the field are being cited among with their results. Explaining anatomy and biomechanics on plain text is extremely difficult to follow. If your team agrees on adding few drawings/sketches for those chapters, I believe this article will bring major value to the scientific community. However, in order to increase our readers pleasure and make it more scientifically sound, I have some suggestions for the authors:
Row 51-53 - can we cite some of those studies?
Row 62 - This chapter (2) clearly requires an anatomic representation (fictive model, cadaveric or graphic designed) Row 63-64 - anterolateral complex (ALC) has been already abbreviated in the Introduction section Row 83-84 - Can we cite some of those several articles? The entire manuscript seems to be translated from a different language and this usually leads to an overpopulated “the” articles troughout the manuscript. I advise removing many of “the” articles altogether with an English revision of phrases. The Chapter 3 has very interesting points with clearly defined biomechanical issues. However, I would suggest to add a figure/sketch/drawing representing this for easier understanding. Row 148-149 - can we sum up the results of the mentioned studies by adding a table/sketch? Row 172-179 - I would also suggest adding a table with those results from different authors
Author Response
This review is a well-written and novel documented on the topic of anterolateral ligament reconstruction and not only. The references of the article are new and the experts in the field are being cited among with their results. Explaining anatomy and biomechanics on plain text is extremely difficult to follow. If your team agrees on adding few drawings/sketches for those chapters, I believe this article will bring major value to the scientific community. However, in order to increase our readers pleasure and make it more scientifically sound, I have some suggestions for the authors:
Author response) Thank you for your comments. We revised the manuscript according to your comments and suggestions. Please see the below.
Row 51-53 - can we cite some of those studies?
Author response) Thank you for the question. We added some references according to your question.
Author action) We added four references regarding the sentence (Line 55 in revised manuscript)
“Patel RM, Brophy RH. Anterolateral Ligament of the Knee: Anatomy, Function, Imaging, and Treatment. Am J Sports Med. 2018 Jan;46(1):217-223”
“Kraeutler MJ, Welton KL, Chahla J, LaPrade RF, McCarty EC. Current Concepts of the Anterolateral Ligament of the Knee: Anatomy, Biomechanics, and Reconstruction. Am J Sports Med. 2018 Apr;46(5):1235-1242”
“Williams A, Ball S, Stephen J, White N, Jones M, Amis A. The scientific rationale for lateral tenodesis augmentation of intra-articular ACL reconstruction using a modified 'Lemaire' procedure. Knee Surg Sports Traumatol Arthrosc. 2017 Apr;25(4):1339-1344”
“Ahn JH, Patel NA, Lin CC, Lee TQ. The anterolateral ligament of the knee joint: a review of the anatomy, biomechanics, and anterolateral ligament surgery. Knee Surg Relat Res. 2019 Nov 28;31(1):12”
Row 62 - This chapter (2) clearly requires an anatomic representation (fictive model, cadaveric or graphic designed)
Author response) Thank you for the comment. We added a figure regarding the anatomy of the ALL according to your comment.
Author action) We added a figure regarding the anatomy of the ALL according to your comment (Figure 1 in revised manuscript).
Row 63-64 - anterolateral complex (ALC) has been already abbreviated in the Introduction section.
Author response) Thank you for the comment. We revised it.
Author action) We removed “an anterolateral complex” as it had been already abbreviated in the Introduction section (Line 67 in revised manuscript)
“The anterolateral aspect of the knee joint consists of multiple structures which constitute the ALC, usually responsible for anterolateral stability.”
Row 83-84 - Can we cite some of those several articles?
Author response) Thank you for the question. We added some references according to your question.
Author action) We added five references regarding the sentence (Line 93 in revised manuscript)
“Nitri M, Rasmussen MT, Williams BT, Moulton SG, Cruz RS, Dornan GJ, Goldsmith MT, LaPrade RF. An In Vitro Robotic Assessment of the Anterolateral Ligament, Part 2: Anterolateral Ligament Reconstruction Combined With Anterior Cruciate Ligament Reconstruction. Am J Sports Med. 2016 Mar;44(3):593-601.”
“Marshall T, Oak SR, Subhas N, Polster J, Winalski C, Spindler KP. Can the Anterolateral Ligament Be Reliably Identified in Anterior Cruciate Ligament-Intact and Anterior Cruciate Ligament-Injured Knees on 3-T Magnetic Resonance Imaging? Orthop J Sports Med. 2018 Sep 24;6(9):2325967118796452.”
“Rasmussen MT, Nitri M, Williams BT, Moulton SG, Cruz RS, Dornan GJ, Goldsmith MT, LaPrade RF. An In Vitro Robotic Assessment of the Anterolateral Ligament, Part 1: Secondary Role of the Anterolateral Ligament in the Setting of an Anterior Cruciate Ligament Injury. Am J Sports Med. 2016 Mar;44(3):585-92.”
“Sonnery-Cottet B, Lutz C, Daggett M, Dalmay F, Freychet B, Niglis L, Imbert P. The Involvement of the Anterolateral Ligament in Rotational Control of the Knee. Am J Sports Med. 2016 May;44(5):1209-14.”
“Kang KT, Koh YG, Park KM, Choi CH, Jung M, Shin J, Kim SH. The anterolateral ligament is a secondary stabilizer in the knee joint: A validated computational model of the biomechanical effects of a deficient anterior cruciate ligament and anterolateral ligament on knee joint kinematics. Bone Joint Res. 2019 Dec 3;8(11):509-517.”
The entire manuscript seems to be translated from a different language and this usually leads to an overpopulated “the” articles throughout the manuscript. I advise removing many of “the” articles altogether with an English revision of phrases.
Author response) Thank you for thoughtful review and comment.
Author action) The entire manuscript has been edited for language again a professional English language editing company (Editage) according to your comments (Please see the entire revised manuscript)
The Chapter 3 has very interesting points with clearly defined biomechanical issues. However, I would suggest to add a figure/sketch/drawing representing this for easier understanding.
Author response) Thank you for the comments. We added a figure and a table for easier understanding.
Author action) We added a figure and a table for easier understanding according to your comment (Table 1 & Figure 2 in revised manuscript).
Row 148-149 - can we sum up the results of the mentioned studies by adding a table/sketch?
Author response) Thank you for the question. We added a table.
Author action) We added a table for easier understanding according to your suggestion (Table 2 in revised manuscript).
Row 172-179 - I would also suggest adding a table with those results from different authors.
Author response) Thank you for the suggestion. We included the contents in Table 1.
Author action) The relevant contents were included in Table 1 for easier understanding according to your suggestion (Table 1 in revised manuscript).
Round 2
Reviewer 2 Report
I would like to congratulate the team for adding increased value to their first form of the manuscript. It is now a great experience to go through your paper and tables, images and work that you've done to improve it will definitely please our readers.